# Heavily and fully modified RNAs guide efficient SpyCas9-mediated genome editing

Aamir Mir [1], Julia F. Alterman[1], Matthew R. Hassler[1], Alexandre J. Debacker[1], Edward Hudgens[2],
Dimas Echeverria[1], Michael H. Brodsky[2], Anastasia Khvorova[1,3], Jonathan K. Watts[1,4] & Erik J. Sontheimer [1,3]

RNA-based drugs depend on chemical modifications to increase potency and to decrease immunogenicity in vivo. Chemical modification will likely improve the guide RNAs involved in CRISPR-Cas9-based therapeutics as well. Cas9 orthologs are RNA-guided microbial effectors that cleave DNA. Here, we explore chemical modifications at all positions of the crRNA guide and tracrRNA cofactor. We identify several heavily modified versions of crRNA and tracrRNA that are more potent than their unmodified counterparts. In addition, we describe fully chemically modified crRNAs and tracrRNAs (containing no 2′-OH groups) that are functional in human cells. These designs will contribute to Cas9-based therapeutics since heavily modified RNAs tend to be more stable in vivo (thus increasing potency). We anticipate that our designs will improve the use of Cas9 via RNP and mRNA delivery for in vivo and ex vivo purposes.

[1] RNA Therapeutics Institute, University of Massachusetts Medical School, Worcester, MA 01605, USA. [2] Department of Molecular, Cell and Cancer Biology, University of Massachusetts Medical School, Worcester, MA 01605, USA. [3] Program in Molecular Medicine, University of Massachusetts Medical School, Worcester, MA 01605, USA. [4] Department of Biochemistry and Molecular Pharmacology, University of Massachusetts Medical School, Worcester, MA 01605, USA. Correspondence and requests for materials should be addressed to A.K. (email: anastasia.khvorova@umassmed.edu) or to J.K.W. (email: jonathan.watts@umassmed.edu) or to E.J.S. (email: erik.sontheimer@umassmed.edu)

CRISPR RNA-guided genome engineering has revolutionized research into human genetic disease and many other aspects of biology. Numerous CRISPR-based in vivo or ex vivo genome editing therapies are nearing clinical trials. At the heart of this revolution are the microbial effector proteins found in class II CRISPR-Cas systems[1] such as Cas9 (type II) and Cas12a/Cpf1 (type V)[2–4].

The most widely used genome editing tool is the type II-A Cas9 from *Streptococcus pyogenes* strain SF370 (SpyCas9)[2]. Cas9 forms a ribonucleoprotein (RNP) complex with a CRISPR RNA (crRNA) and a trans-activating crRNA (tracrRNA) for efficient DNA cleavage both in bacteria and eukaryotes (Fig. 1a). The crRNA contains a guide sequence that directs the Cas9 RNP to a specific locus via base pairing with the target DNA to form an R-loop. This process requires the prior recognition of a protospacer adjacent motif (PAM), which for SpyCas9 is NGG. R-loop formation activates the His–Asn–His (HNH) and RuvC-like endonuclease domains that cleave the target strand and the non-target strand of the DNA, respectively, resulting in a double-strand break (DSB).

For mammalian applications, Cas9 and its guide RNAs can be expressed from DNA (e.g., a viral vector), RNA (e.g., Cas9 mRNA plus guide RNAs in a lipid nanoparticle), or introduced as an RNP. Viral delivery of Cas9 results in efficient editing, but can be problematic because long-term expression of Cas9 and its guides can result in off-target editing, and viral vectors can elicit strong host immune responses[5]. RNA and RNP delivery platforms of Cas9 are suitable alternatives to viral vectors for many applications and have recently been shown to be effective genome editing tools in vivo[6,7]. RNP delivery of Cas9 also bypasses the requirement for Cas9 expression, leading to faster editing. Furthermore,

Cas9 delivered as mRNA or RNP exists only transiently in cells and therefore exhibits reduced off-target editing. For instance, Cas9 RNPs were successfully used to correct hypertrophic cardiomyopathy (HCM) in human embryos without measurable off-target effects[8].

The versatility of Cas9 for genome editing derives from its RNA-guided nature. The crRNA of SpyCas9 used in this study consists of a 20-nt guide region followed by a 16-nt repeat region (Fig. 1a). The tracrRNA consists of an anti-repeat region that pairs with the crRNA, and also includes three stem-loops. All of these secondary structure elements are required for efficient editing in mammalian systems[9]. However, unmodified RNAs are subject to rapid degradation in circulation and within cells[10,11]. Therefore, it is highly desirable to chemically protect RNAs for efficient genomic editing in hard-to-transfect cells and in vivo. Thus, it has been previously reported that chemical modifications in the crRNA and tracrRNA enhance stability and editing efficiency in vivo and ex vivo[6,7,11–13]. Chemical modifications including 2′-O-methyl (2′-OMe), phosphorothioate (PS), 2′-O-methyl thioPACE (MSP), 2′-O-methyl-PACE (MP), 2′-fluoro RNA (2′-F-RNA), and constrained ethyl (S-cEt) have previously been employed to synthesize crRNA and tracrRNA[6,11,12]. The modified RNAs not only improved Cas9 efficacy, but in some instances also improved specificity[11,14]. The effect of an individual modification varies based on the position and combination of chemical modifications used as well as the inter- and intra-molecular interactions with other modified nucleotides. For instance, S-cEt was primarily used to improve oligonucleotide intramolecular folding. Modifications were either based on the crystal structures of Cas9 or limited to the ends of RNAs, and the guides were not modified extensively. Nonetheless, heavily or

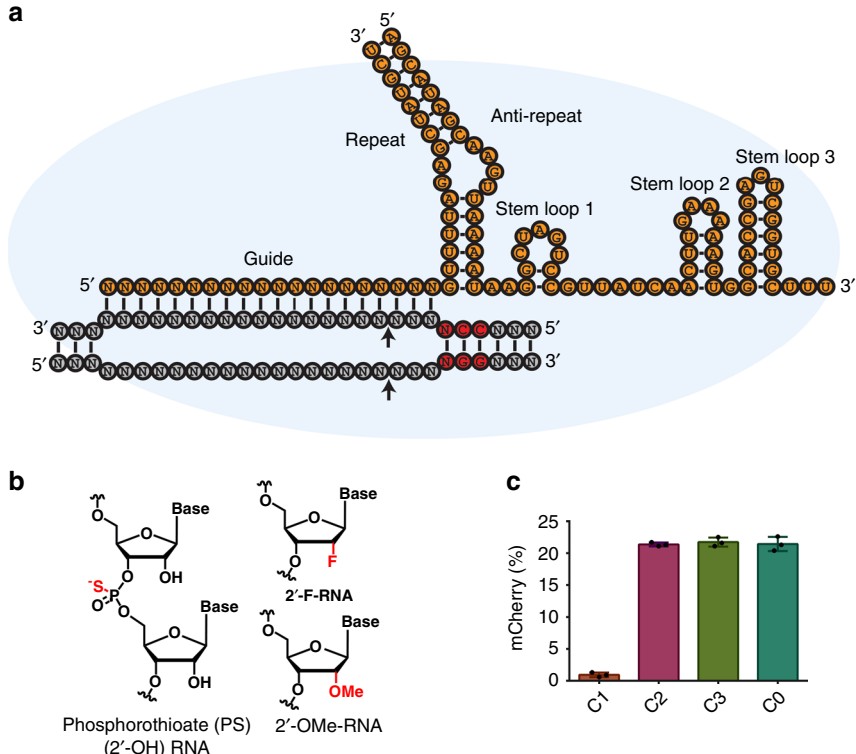

**Fig. 1** Initial screening of chemical modifications in the crRNA. **a** Schematic of Cas9 RNP paired with target DNA. The secondary structure elements of crRNA and tracrRNA are labeled. RNA is shown in orange, whereas DNA is in gray. The PAM sequence is highlighted red and cleavage sites are marked with arrows. **b** Chemical modifications used in this study. **c** Bar graph showing mCherry-positive cells after electroporation of HEK293T-TLR cells with RNPs that included the indicated crRNAs and an unmodified tracrRNA. Error bars represent standard deviation (SD) resulting from at least three biological replicates

**Table 1 Chemically modified crRNAs used in this study**

| | Guide | Repeat |
|---|---|---|
| C1 | GGUGAGCUCUuauuuGCGuA | GuuuUAGAGCUAUGCU |
| C2 | ggugagCUCUuauuugcguA | GuuuuaGAGCUAUGCU |
| C3 | ggugagCUCUuauuugcguA | GuuuuaGAGCUAUGCU |
| C4 | GGUGAGCUCUuauuugcguA | GuuuuaGAGCUAUGCU |
| C5 | ggugagCUCUuauuugCGuA | GuuuuaGAGCUAUGCU |
| C6 | ggugagCUCUuauuugcguA | GuuuUAGAGCUAUGCU |
| C7 | GGUGAGCUCUuauuugCGuA | GuuuuaGAGCUAUGCU |
| C8 | GGUGAGCUCUuauuugcguA | GuuuUAGAGCUAUGCU |
| C9 | GGUGAGCUCUuauuugCGuA | GuuuUAGAGCUAUGCU |
| C10 | GGUGAGCUCUUAUUugCGuA | GuuuUAGAGCUAUGCU |
| C11 | GGUGAGCUCUUAUUugcguA | GuuuUAGAGCUAUGCU |
| C17 | GGUGAGCUCUUAUUugCGuA | GuuuUaGAGCUAUGCU |
| C18 | GGUGAGCUCUUAUUugCGuA | GuuuUAGAGCUAUGCU |
| C19 | GGUGAGCUCUUAUUugCGuA | GuuuuaGAGCUAUGCU |
| C20 | GGUGAGCUCUUAUUugCGuA | GuuuUAGAGCUAUGCU |
| C21 | GGUGAGCUCUUAUUUGCGUA | GUUUUAGAGCUAUGCU |
| C22 | GGUGAGCUCUUAUUugCGuA | GUuUUAGAGCUAUGCU |

Lowercase: 2′-OH; uppercase and bold: 2′-OMe; uppercase and italicized: 2′-F; underlined: 3′ PS

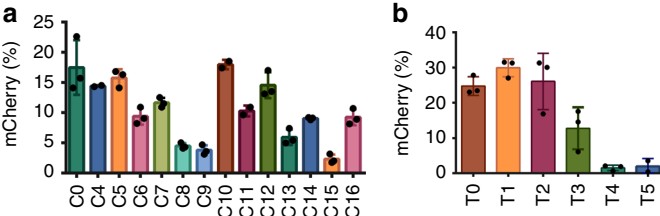

**Fig. 2** Second round of chemical optimization of CRISPR RNAs. The optimized crRNA designs are shown in **a**, whereas chemical designs of tracrRNA are shown in **b**. Each crRNA was tested with the unmodified tracrRNA **T0**, whereas each tracrRNA was tested with the unmodified crRNA **C0**. The bar graphs show the percent of cells expressing mCherry, ±SD. Each RNA was tested in triplicate

fully modified RNAs may have advantages in vivo[10]. Modified siRNAs and ASOs substantially improve stability and potency, and can also reduce off-target effects (though reductions in off-target effects resulting from chemical modifications can be context-dependent)[15]. Furthermore, extensively modified RNAs can prevent innate immune responses[16]. For these and other reasons, the abilities of siRNA and ASO drugs to succeed in clinical trials have hinged on their complete chemical modification and metabolic stabilization, which greatly enhances in vivo efficacy even in cases where in vitro and cell-based activities are decreased[10]. We reasoned that full chemical modification of the crRNA and tracrRNA would likewise enable in vivo therapeutic applications, and set out to identify modification patterns that are compatible with SpyCas9 genome editing function in human cells. We used structure-guided and systematic approaches to introduce 2′-OMe-RNA, 2′-F-RNA, and PS modifications (Fig. 1b) throughout guide RNAs. Our strategy yielded active RNP complexes with both extensively and fully modified versions of crRNAs and tracrRNAs.

## Results

**Structure-based design of crRNA and tracrRNA modifications.** Crystal structures of SpyCas9 have been solved as the RNP alone or bound to one or both strands of target DNA[17–20]. These structures provide detailed information regarding the interactions between the Cas9 protein and crRNA:tracrRNA complex. We used these structures to identify sites where Cas9 protein makes no contacts with the crRNA or tracrRNA. Thus, in our initial screen, 2′-OMe modifications were introduced at guide positions 7–10 and 20 (**C2**, Fig. 1c). Similarly, positions 21 and 27–36 in the crRNA repeat region were also modified using 2′-OMe. To improve nuclease stability, PS modifications were also introduced at the 5′ end of the crRNA, yielding the **C3** design (Fig. 1c and Table 1). In parallel, we tested a crRNA that was more aggressively modified to leave only nine nucleotides (nt) unprotected (**C1**). Similarly, 2′-OMe modifications were also introduced into the tracrRNA at all positions where no protein contact with the RNA is observed. This gave rise to **T1** that is 50% chemically modified (Fig. 2 and Table 2).

The crRNAs and tracrRNAs were tested in a HEK293T cell line stably expressing the traffic light reporter (TLR) system, which includes a GFP (containing an insertion), followed by an out-of-frame mCherry[21]. Upon DSB induction, a subset of non-homologous end-joining (NHEJ) repair events generate indels that place mCherry in frame, leading to red fluorescence. The HEK293T-TLR cells were electroporated (Neon transfection system) with an in vitro-reconstituted RNP complex of recombinant 3xNLS-SpyCas9, crRNA, and tracrRNA. The electroporated cells were analyzed by flow cytometry (Supplementary Fig. 1) for mCherry-positive cells. As shown in Fig. 1c, modified crRNAs **C2** and **C3** retain complete activity relative to the unmodified crRNA **C0**, suggesting that the modifications introduced in crRNAs **C2** and **C3** are well tolerated by Cas9. Lipid-based delivery of the Cas9 RNP complex showed that **C3** is slightly more efficacious than **C0** and **C2** (Supplementary Fig. 2), which is not surprising given the importance of end modifications for Cas9-based genome editing seen previously in other cell types[11]. Since the overall editing efficiency with lipid-based delivery was significantly lower compared to electroporation, we focused on the latter as our preferred mode of Cas9 RNP delivery. Similar to **C2** and **C3**, the modified tracrRNA design **T1** did not hinder Cas9 activity. On the other hand, the extra modifications introduced in **C1** almost completely abolished Cas9 activity in cells. We reasoned that the 2′-OMe modifications (especially at positions 16–18 in the crRNA) are most likely to compromise Cas9 RNP activity since nts at position 16 and 18 were shown to make base-specific contacts with Arg447 and Arg71[17]. The 2′-OH of G16 in the TLR crRNA is also predicted to make a hydrogen bond with Arg447. We chose **C3** and **T1** as a basis for further optimization.

**Table 2 Chemically modified tracrRNAs used in this study**

|  | Anti-repeat | Stem-loop 1 | Linker | Stem-loop 2 | Stem-loop 3 |
|---|---|---|---|---|---|
| T1 | **AGCAUAG**caaguu**A**aaau | aag**G**cu**A**guc**C** | guuauca | **ACUUGAAAAAGU**g | gca**CCG**agucg**GUGCUUU** |
| T2 | **AGCAUAGCAAG**uu**A**a**AA**u | **AAGG**cu**A**guc**C** | guu**AUCA** | **ACUUGAAAAAGUG** | **GCACCGAGUCGGUGCUUU** |
| T3 | **AGCAUAGCAAG**uu**A**a**AA**u | **AAGG**cu**A**guc**C** | guu**AUCA** | **ACUUGAAAAAGUG** | **GCACCGAGUCGGUGCUUU** |
| T4 | **AGCAUAGCAAG**uu**AAA**u | **AAGG**cu**A**guc**C** | guu**AUCA** | **ACUUGAAAAAGUG** | **GCACCGAGUCGGUGCUUU** |
| T6 | **AGCAUAGCAAG**uu**A**a**AA**_U_ | **AAGG**_CUA_g_UC_**C** | guu**AUCA** | **ACUUGAAAAAGUG** | **GCACCGAGUCGGUGCUUU** |
| T7 | **AGCAUAGCAAG**u_UA_a**AA**_U_ | **AAGG**_CUAGUC_**C** | guu**AUCA** | **ACUUGAAAAAGUG** | **GCACCGAGUCGGUGCUUU** |
| T8 | **AGCAUAGCAAG**_UUAAAU_ | **AAGG**_CUAGUC_**C** | _GUU_**AUCA** | **ACUUGAAAAAGUG** | **GCACCGAGUCGGUGCUUU** |

Lowercase: 2′-OH; uppercase and bold: 2′-OMe; uppercase and italicized: 2′-F; underlined: 3′ PS

**Empirical refinement of crRNA and tracrRNA modifications**. In the second round of crRNA modification, we introduced additional 2′-OMe modifications into the first 6 nt of C3 to yield C4 (Fig. 2). In another design, 2′-OMe modifications were incorporated at positions 17 and 18 (C5). G16 was left unmodified because it makes base- and backbone-specific contacts with Cas9 and likely contributed to the low efficacy of C1. Recently, others have also observed similar constraints at position 16[6]. In C6, the importance of 2′-OH groups at positions 25 and 26 was tested. The 2′-OH of these nts contacts the protein in the crystal structure; however, they do not pair with the target DNA, and 2′-OMe substitution at these positions may therefore be more tolerable. C7 and C8 were identical to C5 and C6, respectively, except that they also contained 2′-OMe modifications in the first six positions. All of these crRNAs (C4–C8) were designed to identify modifications responsible for the lower activity of C1 relative to C3.

As shown in Fig. 2 and Supplementary Fig. 3, C4–C7 retain almost the same efficacy as C0, but C8 activity was strongly reduced. These results indicated that nts at positions 1–6 and 17–18 tolerate 2′-OH substitutions. 2′-OMe modifications at positions 25 and 26 were tolerated in C6 but not in C8. In addition, we synthesized a version of C8 that contained PS linkages at several unprotected positions including 15–16, 19, and 21–23 (C9). This design also exhibited reduced editing efficiency by Cas9. When tested for DNA cleavage activity in vitro, C8 and C9 were fully active even at low RNP concentrations (Supplementary Fig. 4). These results suggest the existence of structural perturbations in C8 and C9 that are particularly acute under intracellular conditions.

We also incorporated 2′-F-RNAs in this round of optimization since they can increase thermal and nuclease stability of RNA:RNA or RNA:DNA duplexes, and they also interfere minimally with C3′-endo sugar puckering[22,23]. 2′-F may be better tolerated than 2′-OMe at positions where the 2′-OH is important for RNA:DNA duplex stability. For these reasons, we synthesized two crRNAs based on C9 but with 2′-F modifications at positions 11–14 and/or 17–18 (C10–C11). These modifications rescued some of C9's diminished activity. In fact, C10 (which contained 2′-F substitutions at positions 11–14 and 17–18) performed better than C11, in which positions 17–18 were unmodified. Our results suggest that 2′-F substitutions can compensate for lost efficacy resulting from high 2′-OMe content. However, this may be specific to the modification pattern used in our crRNA design, and the underlying reasons for this compensation remain unclear. It is especially noteworthy that C10 retains the same activity as the unmodified C0 but contains at least one backbone modification at every single phosphodiester linkage. This represents a significant breakthrough for Cas9-based therapeutics because C10 has great potential to provide increased stability, and therefore more efficient editing, in vivo.

We also carried out a second round of tracrRNA optimization. T1 was further modified by introducing 2′-OMe substitutions at most positions where the 2′-OH groups do not make crystal contacts with the protein. In addition, some nts that interact with Cas9 were also modified, given that the crRNA tolerated substitutions at many such positions. This approach produced tracrRNAs T2–T5, which contain modifications in at least 55 out of 67 nts. A15 is the only position that differs between T2 and T4, whereas T3 contains additional stabilizing PS linkages at unprotected positions relative to T2. These tracrRNAs were tested in HEK293T-TLR cells, and the majority of 2′-OMe chemical modifications were tolerated by the tracrRNA except at position A15 (Fig. 2). In the crystal structure, the 2′-OH of A15 interacts with Ser104. The best-performing tracrRNA from this round was T2, which contains 12 unmodified positions. Furthermore, the inclusion of PS linkages at these 12 positions reduced but did not abolish activity. This design (T3) contains at least one chemical modification at every position (either a PS or ribose modification). This also represents an important advance for therapeutic applications of Cas9. It was again interesting to note that while T2 was fully active in cells, it showed lower potency when tested for DNA cleavage in vitro using low concentrations of Cas9 (Supplementary Fig. 4). This discrepancy suggests that net activity in cells and in vitro can be limited by distinct factors or conditions.

The mCherry signal only results from indels producing a +1 frameshift, and therefore underestimates true editing efficiencies. To ensure that crRNA:tracrRNA combinations do not yield false negatives by favoring TLR indels that are out of the mCherry reading frame, we also carried out tracking of indels by decomposition (TIDE) analysis to analyze overall editing efficiencies (Supplementary Figure 3). As shown in Supplementary Fig. 3, editing efficiencies measured using TIDE correlate well with the mCherry signal.

**Terminal conjugates are compatible with modified guides**. We also explored whether addition of terminal modifications such as fluorophores, N-acetylgalactosamine (GalNAc), or cholesterol-triethylene glycol (TEGChol) are tolerated by the crRNA and the tracrRNA. Such modifications can be useful for microscopy, and for monitoring cellular or tissue-specific RNA uptake. We introduced 5′-Cy3 modifications on crRNAs C10 and C11 to yield C12 and C13, respectively (Supplementary Table 1). We also covalently attached TegChol or GalNAc to the 3′ end of C12 or C13 to obtain C14 and C15, respectively. Most crRNA modifications were tolerated on both ends, though some loss of function was observed with C13, C14, and C16 in cells and in vitro (Supplementary Figs. 3 and 4). In contrast, C15 was essentially inactive. T5 containing a 3′-TegChol was also nonfunctional, not surprisingly given the 2′-OMe substitution at A15.

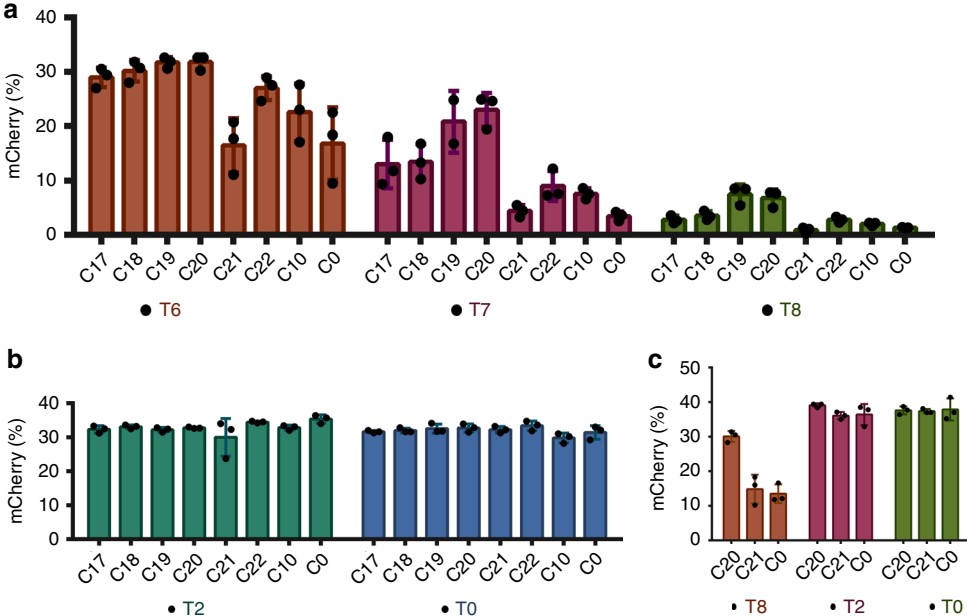

**Fig. 3** Cas9 tolerates heavily and fully modified crRNA:tracrRNA. **a**, **b** Each crRNA was tested with tracrRNAs **T0**, **T2**, and **T6**–**T8** using 20 pmol of Cas9 RNP. **c** HEK293T-TLR cells were also electroporated with 100 pmol of the indicated RNPs to test whether heavily modified RNAs regain functionality at higher doses. Error bars show ± SD of three biological replicates

**Complete chemical modification of dual guides**. We built upon the best-performing individual chemical configurations (**C10** and **T2**) to attempt to define combined crRNA:tracrRNA modification patterns that are compatible with SpyCas9 RNP function. Because crRNA 2′-F substitutions were largely tolerated (Fig. 2), and in some cases even compensated for the loss of efficacy caused by 2′-OMe substitutions, we added several 2′-F modifications to **C10** and **T2**. In addition, because we had observed that simultaneous 2′-OMe modification at positions 25 and 26 negatively affected efficacy in some cases (e.g., **C8**), we tested these two positions for their sensitivities to 2′-F or individual 2′-OMe substitutions. We also incorporated additional 2′-F modifications in the tracrRNAs. In positions where the nucleobases interact with Cas9, we took two approaches to modification. While we suspected that protein-interacting sites would be less tolerant of modification, it was difficult to predict whether steric constraints or charge interactions were more important. To address this issue, we synthesized three different tracrRNAs: one where all protein-interacting sites were left as 2′-OH (**T6**), another where all were converted to 2′-F (**T8**), and another where only the nucleobases that interact with nonpolar amino acids were converted to 2′-F (**T7**). Using this systematic approach, crRNAs **C17**–**C22** and tracrRNAs **T6**–**T8** were synthesized and tested (Fig. 3a).

When **C17**–**C22** were used with either **T2** or the **T0** control (20 pmol RNP), all showed comparable efficacy as the **C0** and **C10** crRNAs (Fig. 3b). This includes the fully modified **C21** that is either 2′-F- or 2′-OMe-substituted at every position. To our knowledge, completely modified and fully functional crRNA has not been reported previously. **C21** loses some efficacy when combined with **T6**–**T8** and is also less potent than **C0** when lower (3 pmole) doses of RNP are delivered (Supplementary Fig. 5). These losses may be due to compromised base pairing between the heavily modified repeat:anti-repeat duplexes. Across all tracrRNAs tested, **C20** exhibits the highest editing efficiency. In addition, at 3 pmol RNP, **C20** is more potent than unmodified **C0**, suggesting enhanced stability in cells (Supplementary Fig. 5). Although **C20** includes six ribose sugars, each is adjacent to a PS modification, leaving no unmodified linkages in the crRNA.

Among **T6**–**T8**, the best-performing tracrRNA was **T6**, especially with modified crRNAs including **C20**. The fully modified tracrRNA (**T8**) compromised the potency of all crRNAs tested but retains some function (~5% editing with 20 pmol RNP) with **C19** and **C20** (Fig. 3b). To test whether **T8** activity improves at higher doses, we electroporated cells with 100 pmol Cas9 RNP. We found that by using a higher amount of Cas9 RNP, the editing efficiency of **T8** in combination with **C0** or **C20** is rescued to the same level as observed using 20 pmol of Cas9 RNP with **C0:T0** (Fig. 3). Furthermore, at higher doses, the efficacy of **C20: T8** is almost as high as that of **C20:T0**. Lastly, the editing efficiency of the fully modified pair (**C21:T8**) is within ~twofold of the unmodified (**C0:T0**) crRNA:tracrRNA pair. To our knowledge, efficient editing with a fully modified crRNA: tracrRNA combination has not been demonstrated previously. While the editing efficiency is not as high as that of the unmodified RNAs in cells, the increased serum stability afforded by the fully chemically optimized **C21:T8** combination (Supplementary Fig. 6) would likely provide significant benefits in vivo, as observed for fully modified siRNAs and ASOs.

**Modified guides support genome editing at endogenous loci**. To verify that our crRNA designs are compatible with different guide sequences, including those targeting endogenous human genes, we tested the **C10**, **C20**, and **C21** designs targeting the huntingtin (*HTT*), human hemoglobin β (*HBB*), and vascular endothelial growth factor A (*VEGFA*) genes[14,24]. *VEGFA* and *HBB* target sites were chosen for their therapeutic potential as well as the fact that they have been previously validated for genome editing. The *HTT* site, on the other hand, is a potential polymorphic target for Huntington's disease treatment. As shown in Fig. 4a, b, *HTT*-**C10** and *HTT*-**C20** performed as well as the minimally modified *HTT*-**C0** when paired with **T2** and **T0**. **T6** and **T7** are more efficacious with the modified **C10** compared to minimally modified **C0**. The fully modified *HTT*-**C21** performed as well as the *HTT*-**C0** when tested with **T2**. However, similar to the TLR target site, some loss of potency is observed with the fully

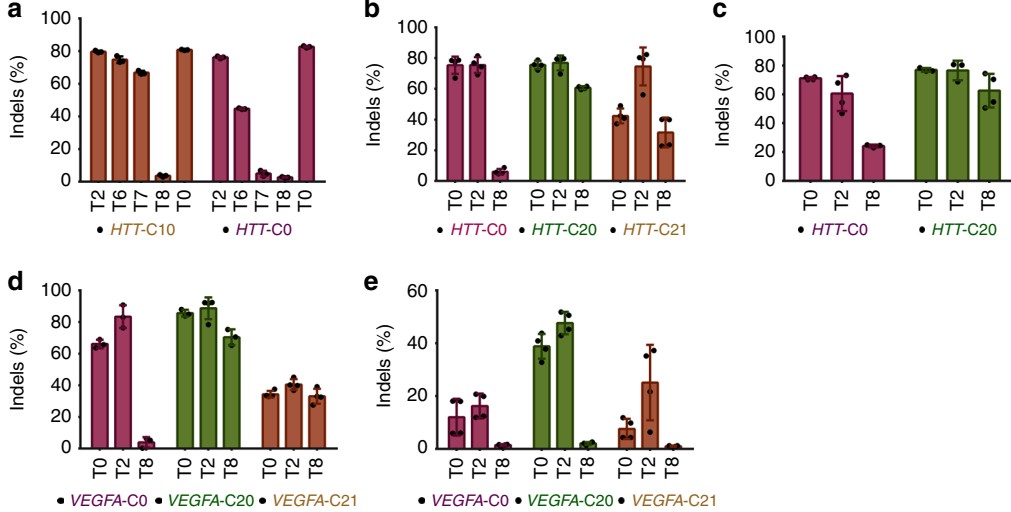

**Fig. 4** Targeting endogenous genes with modified RNAs. **a** The **C10** guide design targeting *HTT* exon 50 was tested using 20 pmol of RNP along with **T2** and **T6**-**T8** in HEK293T cells. **b** *HTT*-**C20** and *HTT*-**C21** designs were tested using 80 pmol of Cas9 RNP with the indicated tracrRNAs. **c** The therapeutically relevant *HBB* locus was targeted using a previously validated guide sequence incorporated into the **C20** design. **d**, **e** *VEGFA*-targeting crRNAs **C20** and **C21** were tested using the indicated tracrRNAs with 80 pmol of RNP in HEK293T cells (**d**) or 60 pmol of RNP in hESCs (**e**). Indels were calculated using TIDE. Bars show averages (±SD) of at least three biological replicates

modified **T8**. However, **T8** did support editing with efficiencies comparable to **T0** when paired with **C20**. Similar results were obtained at the *HBB* and *VEGFA* target sites (Fig. 4c, d): our potent RNA designs (**C20:T2**) performed as well as the minimally modified designs, and the fully modified dual guides exhibited some loss in potency. Furthermore, electroporations performed using 3 pmol of RNP suggested that **C10** and **C20** may be more efficacious (but never less efficacious) than the unmodified crRNA, similar to what was observed in Supplementary Fig. 4, but this effect seemed to vary between target sites (Supplementary Fig. 7). **C20** also showed higher potency compared to **C0** when tested in human embryonic stem cells (hESC) (Fig. 4e). In hESC the highest potency was achieved using the heavily modified combination **C20:T2**. Furthermore, the fully modified crRNA **C21** was just as efficacious as the minimally modified **C0**. We also examined editing in HEK293T cells at the top off-target site for both *HBB* and *VEGFA*, as validated previously[14,24]. The modified crRNAs do not significantly affect off-target editing, though the fully modified **C21:T8** may provide slight specificity improvements compared to the less heavily modified designs (Supplementary Fig. 8). Collectively, these results demonstrate that our modified crRNA designs can be applied to endogenous target sites.

It has previously been shown that crRNA and tracrRNA can be fused with a GAAA tetraloop or other linkers to yield a single-guide RNA (sgRNA) with enhanced efficacy. Given the possibility that repeat:anti-repeat interactions could affect efficacy, we explored the pairing between the repeat and anti-repeat of crRNA and tracrRNA. We designed and synthesized GC-rich crRNAs (**hiGC C1**–**C4**) and tracrRNAs (**hiGC T1**–**T4**) to improve pairing between crRNA and tracrRNA (Supplementary Table 1). All of the modified RNAs outperformed in vitro transcribed sgRNA as well as synthetic, unmodified dual RNAs (Supplementary Fig. 9). Furthermore, at lower concentrations, **hiGC C1** exhibited increased potency relative to non-optimized versions of unmodified or modified RNAs (Supplementary Fig. 9). However, this trend does not hold true in *HTT*-**hiGC C1** (Supplementary Fig. 9). Therefore, these mutant sequences may be superior to wild-type sequences in a guide sequence-specific manner.

**Modified guides support precise editing**. Many genome engineering applications require precise genome editing. For precise repair, a donor DNA can be provided that acts as a template for homology-directed repair (HDR) of a Cas9-mediated DSB. The TLR system[21] allows the use of a donor that restores the functional sequence of the GFP, yielding green fluorescence (NHEJ repairs under these conditions still yield mCherry expression and red fluorescence, so both repair outcomes can be scored simultaneously). To ensure that our modified RNA designs are compatible with HDR, we tested the crRNAs C0, C10, C20, and C21 in combination with tracrRNAs T0, T2, or T8 in HEK293T-TLR cells. In this case, we also provided an 800 bp donor template that contains the GFP sequence (Fig. 5a). Our fully and heavily modified crRNA designs (**C10**, **C20**, and **C21**), when paired with **T0** or **T2**, supported HDR with comparable efficiency as the end-modified **C0** (Fig. 5b). Although **T8** supported HDR at above-background levels (Fig. 5b and Supplementary Data 1), GFP induction was relatively inefficient. This can be attributed to lower overall editing with 20 pmoles of Cas9 RNP (Fig. 5c), as described above. As with NHEJ repairs, HDR activity of the fully modified **C21:T8** was recovered using high dose of Cas9 (Fig. 5d). These experiments demonstrate that our fully and heavily modified RNAs can be used for precise genome editing.

## Discussion

Engineered CRISPR systems have the potential to transform the treatment of inherited diseases via genome editing-based cures. Nonetheless, safe, effective, and target-tissue-specific delivery of CRISPR effector proteins and their small RNA guides represents a major barrier to clinical application. Because of the central importance of the small RNA guides, CRISPR's clinical development could benefit from technologies developed for earlier generations of nucleic acid therapeutics such as siRNAs and antisense oligonucleotides. Two critical realizations have led to a surge of recent successes with these therapeutic modalities: (i) the importance of complete chemical modification (i.e., the removal or modification of 100% of 2′-OH groups) to confer metabolic stability and suppress immune system activation without

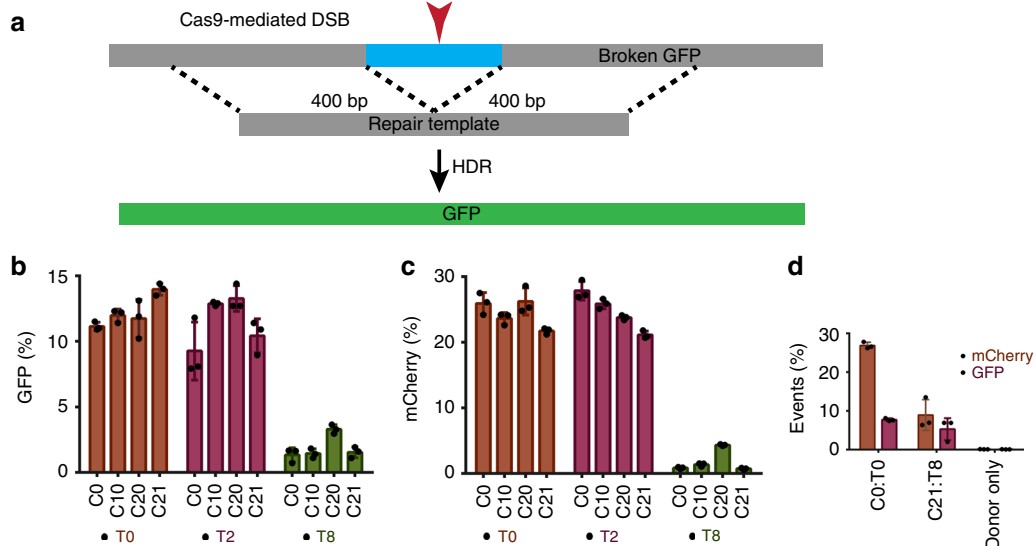

**Fig. 5** Heavily and fully modified RNAs support precise genome editing. **a** Schematic of the Cas9-mediated homology-directed repair in HEK293T-TLR cells. Cells were electroporated with 20 pmoles of Cas9 RNP and 400 ng of 800 bp donor (PCR fragment). Precise repair of the DSB results in GFP expression (**b**), and NHEJ-mediated +1 frameshifts result in mCherry expression (**c**). **d** The fully modified **C21:T8** dual guides were tested using 100 pmoles of Cas9 RNP to test whether fully modified RNAs recover HDR activity at higher doses. Bars show averages (±SD) of three biological replicates

nanoparticle formulation; and (ii) the utility of appended chemical conjugates to tune biodistribution properties and engage cell-surface components that facilitate uptake[10]. These principles should enable the safe, effective delivery of CRISPR guides, either pre-loaded into their protein effectors [ribonucleoprotein (RNP) delivery] or administered in tandem with mRNAs or viral vectors that encode the effector protein.

Here we have used a structure-guided approach, combined with systematic addition of modifications, to identify heavily or fully modified crRNAs and tracrRNAs that direct SpyCas9 genome editing in human cells. Two pairs of crRNA:tracrRNA stand out as particularly promising. First, **C20:T2** is our most potent combination, and both RNAs contain ribose substitutions at >80% of their nts. Furthermore, **C20** contains at least one chemical modification (2′-OMe, 2′-F, or PS) at every single position. The **C20:T2** combination is more potent than its unmodified crRNA:tracrRNA counterpart when tested in human cells. Second, although the **C21:T8** combination exhibits reduced potency in human cells, its significant functionality is still noteworthy because it is completely devoid of ribose sugars. This will greatly ease chemical synthesis, enhance in vivo stability, and provide a springboard toward additional improvements (such as terminally appended chemical functionalities) that facilitate delivery and efficacy during clinical applications of genome editing.

## Methods

**Synthesis of oligonucleotides**. CRISPR guides were synthesized at 1 μmole scale on an Applied Biosystems 394 DNA synthesizer. BTT (0.25 M in acetonitrile, ChemGenes) was used as activator. 0.05 M iodine in pyridine:water (9:1) (TEDIA) was used as oxidizer. DDTT (0.1 M, ChemGenes) was used as sulfurizing agent. A total of 3% TCA in DCM (TEDIA) was used as deblock solution. Oligonucleotides were grown on 1000 Å CPG functionalized with Unylinker (~42 μmol/g). RNA and 2′-OMe phosphoramidites (ChemGenes) were dissolved in acetonitrile to 0.15 M; the coupling time was 10 min for each base. The nucleobase protecting groups were removed with a 3:1 NH₄OH:EtOH solution for 48 h at room temperature or 40% aqueous methylamine for 15 min at 65 °C. Deprotection of the TBDMS group was achieved with DMSO:NEt₃•3HF (4:1) solution (500 μL) at 65 °C for 3 h. RNA oligonucleotides were then recovered by precipitation in 3 M NaOAc (25 μL) and n-BuOH (1 mL), and the pellet was washed with cold 70% EtOH and resuspended in 1 mL RNase-free water.

TracrRNA was synthesized on an Expedite ABI DNA/RNA synthesizer. Sequences were prepared at 1 μmole scale using phosphoramidites (ChemGenes)

prepared as 0.15 M solutions in dry acetonitrile. Trityl groups were removed using 3% dichloroacetic acid (DCA) in toluene for 100 s. All other reagents are the same as above. Deprotection and purification of oligonucleotides were achieved by the addition of 1 mL of 40% aq. methylamine at 60 °C for 13 min. The oligonucleotide solutions were then frozen in liquid nitrogen and lyophilized to dryness in a Speedvac concentrator. Deprotection of the TBDMS group was achieved with DMSO:NEt₃•3HF (4:1) solution (500 μL) at 65 °C for 3 h. The oligonucleotides were then recovered by precipitation in 3 M NaOAc (25 μL) and n-BuOH (1 mL), and the pellet was washed with cold 70% EtOH and resuspended in 1 mL RNase-free water. If the oligo did not contain any 2′TBDMS group, the TBDMS deprotection was omitted.

Purification of oligonucleotides was carried out by high performance liquid chromatography using a 1260 infinity system with an Agilent PL-SAX 1000 Å column (150 × 7.5 mm, 8 μm). Buffer A: 30% acetonitrile in water; Buffer B: 30% acetonitrile in 1 M NaClO₄ (aq). Excess salt was removed with a Sephadex Nap-10 column.

Oligonucleotides were analyzed on an Agilent 6530 Q-TOF LC/MS system with electrospray ionization and time-of-flight ion separation in negative ionization mode. Liquid chromatography was performed using a 2.1 × 50-mm AdvanceBio oligonucleotide column (Agilent Technologies, Santa Clara, CA). The data were analyzed using Agilent Mass Hunter software. Buffer A: 100 mM hexafluoroisopropanol with 9 mM triethylamine in water; Buffer B: 100 mM hexafluoroisopropanol with 9 mM trimethylamine in methanol. Purities are provided in Supplementary Table 3.

**Cell culture**. A human HEK293T stable cell line expressing the traffic light reporter system was kindly provided by Wen Xue's lab in the RNA Therapeutics Institute at UMass Medical School. The original HEK293T cells were obtained from ATCC. These cells were cultured in Dulbecco-modified Eagle's minimum essential medium (DMEM; Life Technologies). DMEM was also supplemented with 10% fetal bovine serum (FBS; Sigma). HEK293T cells were obtained from ATCC and cultured in the same conditions. H1 human embryonic stem cells were obtained from WiCell and cultured using feeder-free mTeSR medium (STEMCELL). Cells were grown in a humidified 37 °C, 5% CO₂ incubator.

**Expression and purification of 3xNLS-SpyCas9**. The pMCSG7 vector expressing the Cas9 from *Streptococcus pyogenes* was kindly provided by Dr. Scot Wolfe's lab. In this construct, the Cas9 also contains three nuclear localization signals (NLSs). Rosetta DE3 strain of *Escherichia coli* was transformed with the 3xNLS-SpyCas9 construct. For expression and purification of 3xNLS-SpyCas9, a previously established protocol was used[2]. The bacterial culture was grown at 37 °C until an OD₆₀₀ of 0.6 was reached. Then, the bacterial culture was cooled to 18 °C, and 1 mM isopropyl β-D-1-thiogalactopyranoside (IPTG; Sigma) was added to induce protein expression. Cells were grown overnight for 16–20 h.

The bacterial cells were harvested and resuspended in lysis buffer [50 mM Tris-HCl (pH 8.0), 5 mM imidazole]. 10 μg/mL of lysozyme (Sigma) was then added to the mixture and incubated for 30 min at 4 °C. This was followed by the addition of

1 × HALT Protease Inhibitor Cocktail (ThermoFisher). The bacterial cells were then sonicated and centrifuged for 30 min at $18,000 \times g$. The supernatant was then subjected to Nickel affinity chromatography (Qiagen). The elution fractions containing the SpyCas9 were then further purified using cation exchange chromatography using a 5 mL HiTrap S HP column (GE). This was followed by a final round of purification by size-exclusion chromatography using a Superdex-200 column (GE). The purified protein was concentrated and flash frozen for subsequent use.

**Electroporations of mammalian cells.** The HEK293T and hESC cells were electroporated using the Neon transfection system (ThermoFisher) according to the manufacturer's protocol. Briefly, 20–100 picomoles of 3xNLS-SpyCas9 were mixed with 25–125 picomoles of crRNA:tracrRNA in buffer R (ThermoFisher) and incubated at room temperature for 20–30 min. For electroporations involving HDR experiments, 400 ng of dsDNA donor was also added to the mixture. This Cas9 complex was then mixed with ~100,000 cells, which were already resuspended in buffer R. This mixture was electroporated with a 10 μL Neon tip and then plated in 24-well plates containing 500 μL of the appropriate media. The cells were stored in a humidified 37 °C and 5% $CO_2$ incubator for 2–3 days for HEK293T and a week for hESCs.

**Flow cytometry.** The electroporated HEK293T cells were analyzed on MACS-Quant® VYB from Miltenyi Biotec. For mCherry detection, the yellow laser (561 nm) was used for excitation and 615/20 nm filter used to detect emission. At least 20,000 events were recorded and the subsequent analysis was performed using FlowJo® v10.4.1. Cells were first sorted based on forward and side scattering (FSC-A vs SSC-A) to eliminate debris (Supplementary Figure S1). Then, cells were gated using FSC-A and FSC-H to select single cells. Finally, mCherry signal was used to select for mCherry-expressing cells. The percent of cells expressing mCherry was calculated and reported in this study as a measure of Cas9-based genome editing.

**Indel analysis by TIDE.** The genomic DNA from cells was harvested using DNeasy Blood and Tissue kit (Qiagen) as recommended by the manufacturer. Approximately 50 ng of genomic DNA was used to PCR amplify a ~700 bp fragment using primers shown in Supplementary Table 2. The PCR reactions were subsequently purified using a QIAquick PCR Purification kit (Qiagen). The PCR fragments were then sequenced by Sanger sequencing and the trace files were subjected to indel analysis using the TIDE web tool (https://tide.deskgen.com/).

**In vitro DNA cleavage assays.** The traffic light reporter plasmid (Addgene: 31482) was linearized with restriction enzyme EcoRI (N.E.B.) for 1 h, 37 °C in NEB buffer 3, followed by heat inactivation for 20 min at 65°C. For the Cas9 digest, 200 ng of linearized plasmid DNA was added to pre-formed RNP complexes (8 pmol or 0.8 pmol) and incubated for 1 h at 37°C in 25 μl NEB buffer 3. Cut DNA was purified using Zymo DNA purification columns and separated on a 1% agarose gel run at 100 V. Relative intensities of full length and Cas9-cut DNA fragments were determined using ImageJ software.

**Serum stability assays.** A 10 μM Cas9 RNP complex was first assembled in cleavage buffer [20 mM HEPES (pH 7.5), 250 mM KCl and 10 mM $MgCl_2$]. Then, 2 μM Cas9 RNP was incubated with 8% FBS in a 50 μL reaction at 37 °C. Then, at time points of 0, 1, and 20 h, 10 μL of the reaction mixture was treated with Proteinase K and then 10 μL of quench buffer (90% formamide and 25 mM EDTA) was added to the solution. The reaction mixture was resolved on a 10% denaturing polyacrylamide gel containing 6 M Urea. The gel was stained with SYBR safe and visualized on Typhoon FLA imager.

**Data availability.** The authors declare that all data supporting the findings of this study are available within the paper and supplementary information. Any other data related to this manuscript are available from the corresponding author on reasonable request.

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

## Acknowledgements

The authors acknowledge partial support from the CHDI Foundation Research Contract A-10199 to M.H.B. We also like to thank Nadia Amrani (Sontheimer lab) for her assistance with culturing hESCs. We are grateful to Scot Wolfe, Wen Xue, and members of their labs for materials and advice, and to all members of the Sontheimer, Watts, Khvorova, and Brodsky labs for helpful discussions.

## Author contributions

All authors participated in crRNA and tracrRNA design; A.M., M.R.H., A.J.D., and D.E. synthesized and purified crRNAs and tracrRNAs; A.M. expressed and purified recombinant SpyCas9; A.M. and E.H. conducted cellular genome editing experiments; A.M., J.F.A., J.K.W., and E.J.S. wrote the manuscript; and all authors edited the manuscript.

## Additional information

**Competing interests:** The authors declare the following competing interests: a patent application has been filed by the University of Massachusetts Medical School describing the inventions reported herein, with the authors as inventors. E.J.S. is a co-founder and Scientific Advisory Board member of Intellia Therapeutics.

