## [Peer Review File · Nature Communications]

Reviewers' comments:

Reviewer #1 (Remarks to the Author):

I would like to comment the authors for their work. The manuscript details a large body of work that is pertinent and should be of broad interest to the field.

I would recommend the publication of the manuscript with some significant revisions.

General Comments:

I. Your statement at the end of the manuscript: "While the editing efficiency is not as high as that of the unmodified RNAs in cells, the increased serum stability afforded by the fully chemically optimized C21:T8 combination (Figure S5) would likely provide significant benefits in vivo, as observed for fully modified siRNAs and ASOs" is actually the thesis or justification for your experiments. This should be clear and stated upfront in your introduction. Currently, siRNAs that are having success in clinical trials are fully modified with the modifications that you use in your guide RNAs; 2'-O-methyl, 2'-F, phosphorothioate. Those siRNAs have greatly diminished activity in cell culture as compared to siRNA duplexes that have been optimized for use in cell culture, it is logical that the same would hold true for guide RNAs. Your current introduction rambles and never gets to that point. Please rewrite the introduction with this in mind.

II. The reader needs to be reassured that the chemically synthesized RNAs that you are making are consistent enough in purity that differences that you see in both in vitro and in cell culture are a result of the modification patterns and not variations in the purity or integrity the various sequences that you have screened. Ideally, I would like to see a table of the oligos with a listing of the full-length purity as determined by LC/MS, ion-pair chromatography, or ion-exchange where you can resolve the 67mer from the 66, 65 etc. Because you are incorporating 2'-F modifications it would be best if this was done by LC/MS and compare the various peak heights from the extracted ion chromatogram assuming consistent ionization of the various length sequences (which is reasonable). You are using methylamine/ammonia to deprotect your RNA and this can lead to a significant amount of depyrimidination on 2'-F U.

III. Wading through your experiments is a bit tedious, part of the issue is having to flip back and forth to the tables of sequence modifications and results in figures. If there is a way to streamline that it would be helpful. Please make sure that you make it clear what experiments were done in cell culture and what were done in vitro. That is important because you don't talk about the relationship between your in vitro results and the cell culture results. I have seen several presentations where the in vitro cleavage results do not match the cell based data. That may be worth a comment for clarity.

Specific Comments:

1. In the abstract the authors should change "RNA-based drugs depend on chemical modifications to increase potency and nuclease stability, and to decrease immunogenicity in vivo." The increase in potency seen with modified nucleosides is not separate from nuclease stability, and some of the "potency" of oligonucleotide therapeutics can be linked to immunogenicity. When you talk about potency separate from nuclease stability or immunogenicity it gives the impression they are different.

2. I am unconvinced that adding lots of mods will improve the use of Cas9 via RNP and mRNA delivery for ex vivo, based upon existing data, exonuclease "end-protection" seems to be enough for ex vivo applications? Do you have additional data that endonucleases are a problem in ex vivo applications? How does it affect HDR?

3. In Hendel, A. et al., 2'-O-methyl didn't show an increase in efficiency over unmodified; most-likely because it doesn't add exonuclease stability.

4. In Ryan, D. et.al., they use the 2'-O-methyl-PACE (MP) to both stabilize the ends from exonucleases and increase the sequence specificity, you should add this to the list.

5. It is hard to generalize about the constrained ethyl (S-cEt) modification because it was used in Rahdar, M. et al., it was used to shorten the duplex region between the crRNA and tracrRNA to a length that would not form a stable duplex with unmodified RNA; it didn't show increased nuclease resistance or specificity? Like 2'-O-methyl it was used in a CRISPR RNA but didn't demonstrate a useful effect on its own. When describing the use of modifications being more specific is better for the reader; specific modifications have specific effects (you generally do a good job of this later in the manuscript).

6. You should be a little more specific when you say Modified siRNAs and ASOs substantially improve stability and potency, and can also reduce off-target effects. For siRNA off-target reduction was demonstrated by one of the authors in Khvorova, et. al. Cell 115(1), 209-216 (2003) using modified RNAs. But this is a unique use of modifications to stabilize one end of the duplex and get preferential loading in RICS. Most of the modifications used in ASOs to date increase off-target effects; once again be more specific.

7. You should say that you transfected your cells by electroporation and not leave it to the trade name of the device.

8. Our results suggest that 2'-F substitutions can compensate for lost efficacy resulting from high 2'-OMe content. When I look at your data I don't think this statement is completely supported. 2'-O-methyl increases T_m over RNA by about 0.2° at physiological salt conditions it doesn't decrease the binding. With the exception of C8, all of the decreased indel measurements you are commenting upon have phosphorothioates starting at position 15 and extending to the UUU in the repeat. I am more likely to believe that the P=S disrupts the duplex than the 2'-O-methyl; maybe give other possible explanations.

9. Once again, I am specifically concerned with the use of 2'-F U in your sequences. 2'-F U imparts endonuclease stability in duplex structures, but is chemically unstable and likes to fall apart to both the urea and the abasic site. It can be managed when only one or two are used but in long RNAs with multiple incorporations it is going to drastically affect the purity and integrity of the resulting RNA.

10. The authors need to be specific when they talk about modifications and the effect that those modifications have on duplex stability, protein contacts to the RNA, and non-specific binding. Modifying RNA can be a double-edged sword; phosphorothioates inhibit nucleases but increase non-specific binding. What about off-targets? A final comment would suffice

Reviewer #2 (Remarks to the Author):

This manuscript describes a study of the activity of chemically modified crRNAs and tracrRNAs. It reports the activity for a number of heavily modified cr and tracrRNAs, including combinations with no remaining 2'-OH groups that retain some activity. Modifications of the RNAs used to guide CRISPR-Cas9 activity could yield much greater stability for in vivo applications.

The problem addressed here is of practical significance for potential in vivo applications of CRISPR. These results provide helpful context for future efforts to obtain high-stability versions of specific crRNA and tracrRNAs intended for therapeutic application.

The finding in this study that a heavily modified crRNA could preserve CRISPR activity is both interesting and encouraging and important to report. It is not clear that Nature Communications is the appropriate venue for this study. The details of which modifications were and were not

tolerated are perhaps more appropriate to a more specialized audience especially given that they are not yet shown to be general properties of *S. pyogenes* crRNAs. The relevance of the reported activity assays to in vivo conditions is not evaluated, and generalizability across crRNA sequences is only minimally examined. With respect to therapeutic application, the effects of the modifications on both activity and stability will need to be determined for the actual crRNA sequences under relevant conditions. The modification patterns that were shown to preserve activity in this study could be quite helpful in suggesting a starting point for developing a modified RNA for therapeutic purposes, and are in that regard quite interesting and useful, but they are still far from providing a validated prescription for predictably achieving the desired properties in vivo.

Major Concerns:

1. The generalizability of the results in this study across different crRNA sequences is very important to the relevance and utility of the findings. This work relies heavily on modifications of a single guide sequence. To support that "crRNA designs are compatible with different guide sequences including those targeting endogenous human genes" one modified form (C10) and the unmodified form of one additional crRNA sequence is tested (paired with several of their previously employed modified tracrRNAs). The activity this second crRNA sequence with the single modification reassuringly shows similar activity to the unmodified oligo, but this is the minimal amount level of testing of generalizability that could be done. Either the reliance on a single crRNA for most of the activity results shown should be more clearly acknowledged in the text or tests of more of the modifications on more crRNA sequences must be done.

2. On p. 4, the statement is made that "Lipid-based delivery of Cas9 RNP complex showed that C3 was even more efficacious than C0 and C2 (Figure S1), which demonstrates the importance of PS linkages at the 5'terminus of the crRNA". Figure S1 shows the stated difference of C3 versus C0 and C2, in stark contrast to Fig. 1C which shows no difference, and also shows much higher mCherry frameshift rates overall than Fig. S1. If lipid-based delivery shows such substantial differences versus nucleofection delivery with respect in activity rates measured by the mCherry assay - both in absolute levels and relative levels between different crRNA forms, then which is trustworthy? Most of the data in the paper used nucleofection; is this reliable? If so, why trust the result in Fig. 1S which disagree with the nucleofection result. Why does delivery affect relative activity? And since it does, how can one know which delivery conditions provide the relevant answers with respect to activity?

3. The text should be more careful not to draw poorly supported conclusions about which modifications matter in a general way based on one or two modification variations on a single oligo sequence. The narrative often gives rationales (quite sensible ones) for why the authors thought certain modifications would be tolerated, but getting the expected result for a single variant or two provides very thin support that rationale was correct.

4. Why aren't the results for C1 and C3 shown in Fig. 2A? They were not re-done in this experiment vs. the one shown in Fig. 1C? The results for C0 is a bit different quantitatively in Fig. 1C and Fig. 2A. What factors governed the reproducibility of the mCherry signal from the TLR system that this study relies so heavily on?

Mir et al., NCOMMS-18-05832-T ("Heavily and fully modified RNAs guide efficient SpyCas9-mediated genome editing")

Response to Critiques

We are very grateful to the reviewers for their time and insight into our manuscript and the work that it described. The comments and criticisms (reproduced in red below) have enabled us to improve the manuscript significantly. Our responses are detailed below, and the corresponding changes made to the manuscript are highlighted in yellow.

Reviewers' comments:

Reviewer #1 (Remarks to the Author):

I would like to comment the authors for their work. The manuscript details a large body of work that is pertinent and should be of broad interest to the field.

I would recommend the publication of the manuscript with some significant revisions.

Response: We are extremely grateful to the reviewer for such positive feedback regarding our manuscript. We are glad that the reviewer shares our excitement regarding the broad applicability of our findings for the field. We also appreciate the highly constructive criticisms below.

General Comments:

I. Your statement at the end of the manuscript: "While the editing efficiency is not as high as that of the unmodified RNAs in cells, the increased serum stability afforded by the fully chemically optimized C21:T8 combination (Figure S5) would likely provide significant benefits in vivo, as observed for fully modified siRNAs and ASOs" is actually the thesis or justification for your experiments. This should be clear and stated upfront in your introduction. Currently, siRNAs that are having success in clinical trials are fully modified with the modifications that you use in your guide RNAs; 2'-O-methyl, 2'-F, phosphorothioate. Those siRNAs have greatly diminished activity in cell culture as compared to siRNA duplexes that have been optimized for use in cell culture, it is logical that the same would hold true for guide RNAs. Your current introduction rambles and never gets to that point. Please rewrite the introduction with this in mind.

Response: We are thankful to the reviewer for recognizing the importance of and advocating for the fully modified guide RNAs. We have rewritten parts of the introduction to stress the utility of fully modified RNAs for therapeutic purposes.

II. The reader needs to be reassured that the chemically synthesized RNAs that you are making are consistent enough in purity that differences that you see in both in vitro and in cell culture are a result of the modification patterns and not variations in the purity or integrity the various sequences that you have screened. Ideally, I would like to see a table of the oligos with a listing of the full-length purity as determined by LC/MS, ion-pair chromatography, or ion-exchange where you can resolve the 67mer from the 66, 65 etc. Because you are incorporating 2'-F modifications it would be best if this was done by LC/MS and compare the various peak heights from the extracted ion chromatogram assuming consistent ionization of the various length sequences (which is reasonable). You are using methylamine/ammonia to deprotect your RNA and this can lead to a significant amount of depyrimidination on 2'-F U.

Response: We thank the reviewer for raising this important point. We take the quality control of our compounds very seriously and have now included the LC/MS chromatograms and deconvoluted mass spectrum in the supporting information confirming the M- is the expected product. Additionally, compounds with this modification are regularly used in clinical candidates and have not shown a de-pyrimidination issue.

Furthermore, in previous work we have carried out on 2'-fluorinated oligonucleotides (PMID 19590787), we did not see significant degradation with the conditions used for deprotection of these compounds (we tested methylamine up to several days). We did see some degradation under more strongly basic conditions (e.g. 1M sodium hydroxide at 65 °C for 8-20h).

III. Wading through your experiments is a bit tedious, part of the issue is having to flip back and forth to the tables of sequence modifications and results in figures. If there is a way to streamline that it would be helpful. Please make sure that you make it clear what experiments were done in cell culture and what were done in vitro. That is important because you don't talk about the relationship between your in vitro results and the cell culture results. I have seen several presentations where the in vitro cleavage results do not match the cell-based data. That may be worth a comment for clarity.

Response: We agree that the *in vitro* data does not always match cell-based data, though based on other systems involving RNA modifications (e.g. siRNAs), we do not consider this surprising. In our case, the discrepancies are relatively few, and of modest magnitude. Our *in vitro* assays, for the most part, are consistent with the cell-based assays. We devoted text to clearly discuss the discrepancies between *in vitro* and cell-based assays, and to specify which experiments were *in vitro* vs. in cells.

Specific Comments:

1. In the abstract the authors should change “RNA-based drugs depend on chemical modifications to increase potency and nuclease stability, and to decrease immunogenicity in vivo.” The increase in potency seen with modified nucleosides is not separate from nuclease stability, and some of the “potency” of oligonucleotide therapeutics can be linked to immunogenicity. When you talk about potency separate from nuclease stability or immunogenicity it gives the impression they are different.

Response: We agree that *in vivo* potency and nuclease stability are often interdependent. However, there are clear examples where chemical modification of oligonucleotide drugs increases their potency in ways beyond nuclease stability. (For one clear direct comparison across many sequences, see PMID 26578588 by Lennox and Behlke – in that case, ASO potency drops significantly from LNA gapmers to 2'OMeRNA gapmers to PS DNA ASOs, in spite of the fact that all the ASO backbones are fully PS-modified and thus relatively nuclease stable). Nonetheless, we have clarified the wording of the abstract accordingly.

2. I am unconvinced that adding lots of mods will improve the use of Cas9 via RNP and mRNA delivery for ex vivo, based upon existing data, exonuclease “end-protection” seems to be enough for ex vivo applications? Do you have additional data that endonucleases are a problem in ex vivo applications? How does it affect HDR?

Response: The increase in potency seen in C20:T2 (Figure S4) relative to C0:T0 may be indicative of increased nuclease protection offered by the heavily modified RNAs. Furthermore, we have included new data where we electroporated hESC and found that our heavily modified designs show increased genome editing efficiencies relative to end-modified RNAs (Figure 4E). It is known that cells of hematopoietic

origin require protective modifications in RNAs for efficient genome editing (e.g. PMID 26121415). We are not certain if this is due to increased nuclease stability or some other reason.

As far as HDR is concerned, we think that HDR rates are mostly dependent on the efficacy of the RNAs for genome editing and donor availability. We expect the HDR rates to be correlated to the overall efficacy of modified RNAs as seen in Hendel *et al.* (2015). We have added an additional figure in the main text (Figure 5) which shows that crRNAs C10, C20 and C21 support HDR to the same extent as C0. On the other hand, the efficacy of T2 for HDR is the same as the end-modified T0, whereas T8 shows some loss in efficacy due to lower potency for overall genome editing.

3. In Hendel, A. et al., 2'-O-methyl didn't show an increase in efficiency over unmodified; most-likely because it doesn't add exonuclease stability.

Response: Based on our examination of the data in Hendel *et al.*, we would argue that 2'-O-methyl end modification did improve the editing efficiency, albeit modestly, compared to unmodified crRNA. (It is certainly true that the effects of certain other modifications were more dramatic).

The 2'-OMe modification does offer protection from endonucleases, and all of our designs also include phosphorothioate modifications at the termini which contributes to exonuclease stability.

4. In Ryan, D. et al., they use the 2'-O-methyl-PACE (MP) to both stabilize the ends from exonucleases and increase the sequence specificity, you should add this to the list.

Response: Good suggestion – done.

5. It is hard to generalize about the constrained ethyl (S-cEt) modification because it was used in Rahdar, M. et al., it was used to shorten the duplex region between the crRNA and tracrRNA to a length that would not form a stable duplex with unmodified RNA; it didn't show increased nuclease resistance or specificity? Like 2'-O-methyl it was used in a CRISPR RNA but didn't demonstrate a useful effect on its own. When describing the use of modifications being more specific is better for the reader; specific modifications have specific effects (you generally do a good job of this later in the manuscript).

Response: We thank the reviewer for these useful insights into the effect of individual chemical modifications. We have now included brief text (lines 43-45) regarding the effect of different modifications. Rahdar *et al.* used S-cEt modification on the 3' end of crRNA to improve the pairing between crRNA and tracrRNA. When used with other modifications it did show an increase in potency both in short and long crRNA designs. S-cEt does improve nuclease stability as this has been shown directly in other studies (Seth *et al.* – PMID 20136157).

6. You should be a little more specific when you say Modified siRNAs and ASOs substantially improve stability and potency, and can also reduce off-target effects. For siRNA off-target reduction was demonstrated by one of the authors in Khvorova, et. al. Cell 115(1), 209-216 (2003) using modified RNAs. But this is a unique use of modifications to stabilize one end of the duplex and get preferential loading in RICS. Most of the modifications used in ASOs to date increase off-target effects; once again be more specific.

Response: We agree with the reviewer that chemical modifications can increase or decrease off-target effects based on the pattern of modifications and sequence of oligo used. We have added this to the text (lines 48-50). We are grateful to the reviewer for pointing this out.

7. You should say that you transfected your cells by electroporation and not leave it to the trade name of the device.

Response: This is an excellent point. We have corrected the manuscript according to this suggestion.

8. “Our results suggest that 2'-F substitutions can compensate for lost efficacy resulting from high 2'-OMe content.” When I look at your data I don't think this statement is completely supported. 2'-O-methyl increases T_m over RNA by about 0.2° at physiological salt conditions it doesn't decrease the binding. With the exception of C8, all of the decreased indel measurements you are commenting upon have phosphorothioates starting at position 15 and extending to the UUU in the repeat. I am more likely to believe that the P=S disrupts the duplex than the 2'-O-methyl; maybe give other possible explanations.

Response: We thank the reviewer for this observation. By comparing the crRNAs C8 and C9, we do not believe that the loss in efficacy in C9 crRNA is due to PS linkages (these two sequences have similar activity and different PS content).

The reviewer is correct that 2'OMeRNA does not show decreased binding affinity to complementary RNA or DNA (relative to native RNA), and we didn't intend to imply that. There appear to be other steric or structural interactions at play here (whether interactions with the protein, the global helical structure, etc.) that appear to disfavor Cas9 activity when those residues are 2'OMe-RNA but not when they are 2'F-RNA. We have clarified our text in this section to avoid making an incorrect implication.

9. Once again, I am specifically concerned with the use of 2'-F U in your sequences. 2'-F U imparts endonuclease stability in duplex structures, but is chemically unstable and likes to fall apart to both the urea and the abasic site. It can be managed when only one or two are used but in long RNAs with multiple incorporations it is going to drastically affect the purity and integrity of the resulting RNA.

Response: See answer from above (Reviewer #1's general comment II).

10. The authors need to be specific when they talk about modifications and the effect that those modifications have on duplex stability, protein contacts to the RNA, and non-specific binding. Modifying RNA can be a double-edged sword; phosphorothioates inhibit nucleases but increase non-specific binding. What about off-targets? A final comment would suffice.

Response: We have added a final comment as the reviewer suggests to discuss the possible outcomes of chemical modification. We agree that modifications on crRNA can increase off-target effects; however, the protein binding of PS linkages may not lead to off-target genome editing. To explore this, we tested the top off-target site of *VEGFA* and *HBB* genomic sites for **C0:T0**, **C20:T2** and/or **C21:T8**. At the concentration tested, the off-target editing either remained the same as C0:T0 or in some cases (**C21:T8**) specificity may have improved.

Reviewer #2 (Remarks to the Author):

This manuscript describes a study of the activity of chemically modified crRNAs and tracrRNAs. It reports the activity for a number of heavily modified cr and tracrRNAs, including combinations with no remaining 2'-OH groups that retain some activity. Modifications of the RNAs used to guide CRISPR-Cas9 activity could yield much greater stability for in vivo applications.

The problem addressed here is of practical significance for potential *in vivo* applications of CRISPR. These results provide helpful context for future efforts to obtain high-stability versions of specific crRNA and tracrRNAs intended for therapeutic application.

The finding in this study that a heavily modified crRNA could preserve CRISPR activity is both interesting and encouraging and important to report. It is not clear that Nature Communications is the appropriate venue for this study. The details of which modifications were and were not tolerated are perhaps more appropriate to a more specialized audience especially given that they are not yet shown to be general properties of *S. pyogenes* crRNAs. The relevance of the reported activity assays to *in vivo* conditions is not evaluated, and generalizability across crRNA sequences is only minimally examined. With respect to therapeutic application, the effects of the modifications on both activity and stability will need to be determined for the actual crRNA sequences under relevant conditions. The modification patterns that were shown to preserve activity in this study could be quite helpful in suggesting a starting point for developing a modified RNA for therapeutic purposes, and are in that regard quite interesting and useful, but they are still far from providing a validated prescription for predictably achieving the desired properties *in vivo*.

Response: We thank the reviewer for this thoughtful critique. In response to concerns about generalizability of multiple modification patterns to additional sequences, we have addressed this in several ways. First, we tested additional guide sequences targeting *HTT*, *HBB* and *VEGFA* loci, in the context of the **C0**, **C10**, **C20** and **C21** designs. In the original submission, we had only included tests of *HTT*, and in that case, only with the C10 modification framework. Our conclusions based on TLR targeting were largely recapitulated at each of these endogenous sites. Our extension of our results into several modification patterns across three completely distinct target sequences greatly improves our confidence that our results apply more broadly than just the TLR. Furthermore, we tested our most highly modified *VEGFA* guides in another cell line where chemical modifications are likely to be more useful, i.e. hESCs. In this case the T2:C20 modification had an even more dramatically positive effect on editing than in HEK293T cells. Overall our results considerably extend those from our initial manuscript. We are grateful to the reviewer for helping us to improve the work.

Regarding the comments about the effects of heavy or complete modification *in vivo* or in therapeutic applications, we fully agree that this will be a critically important point to examine, and in fact that has been our longer-term goal since we began our study. We have every intention of addressing these questions in future manuscripts. The primary message of this manuscript is that full RNA modification is in fact attainable for SpyCas9 editing in an intracellular setting. We consider this to be a crucial advance in and of itself and that the fast-moving CRISPR genome editing field should be able to capitalize on this framework sooner rather than later. Accordingly, to avoid having *in vivo* analyses unduly compromise the timeliness of our initial report, we are opting to defer animal experiments to future manuscripts.

Major Concerns:

1. The generalizability of the results in this study across different crRNA sequences is very important to the relevance and utility of the findings. This work relies heavily on modifications of a single guide sequence. To support that "crRNA designs are compatible with different guide sequences including those targeting endogenous human genes" one modified form (C10) and the unmodified form of one additional crRNA sequence is tested (paired with several of their previously employed modified tracrRNAs). The activity this second crRNA sequence with the single modification reassuringly shows similar activity to the unmodified oligo, but this is the minimal amount level of testing of generalizability that could be done.

Either the reliance on a single crRNA for most of the activity results shown should be more clearly acknowledged in the text or tests of more of the modifications on more crRNA sequences must be done.

Response: Please see our response to Reviewer #2's opening comments.

2. On p. 4, the statement is made that "Lipid-based delivery of Cas9 RNP complex showed that C3 was even more efficacious than C0 and C2 (Figure S1), which demonstrates the importance of PS linkages at the 5' terminus of the crRNA". Figure S1 shows the stated difference of C3 versus C0 and C2, in stark contrast to Fig. 1C which shows no difference, and also shows much higher mCherry frameshift rates overall than Fig. S1. If lipid-based delivery shows such substantial differences versus nucleofection delivery with respect in activity rates measured by the mCherry assay - both in absolute levels and relative levels between different crRNA forms, then which is trustworthy? Most of the data in the paper used nucleofection; is this reliable? If so, why trust the result in Fig. 1S which disagree with the nucleofection result. Why does delivery affect relative activity? And since it does, how can one know which delivery conditions provide the relevant answers with respect to activity?

Response: It is entirely possible that one delivery mode vs. another (e.g., endosomal vs. non-endosomal) is differentially sensitive to modification patterns. Thus, when (modest) relative differences in modification pattern efficacy are observed with different delivery systems, we do not agree that one must be deemed "trustworthy" or "reliable," "provide relevant answers," etc. and the other not – they are different, that is all. We focused most of the manuscript on RNP electroporation for two primary, related reasons: (i) this delivery method gives the greatest efficiency of editing, and (ii) this method is by far the preferred RNP delivery route by the CRISPR genome editing field in general. These considerations maximize the utility of our electroporation results for the rest of the field, to the benefit of the manuscript.

3. The text should be more careful not to draw poorly supported conclusions about which modifications matter in a general way based on one or two modification variations on a single oligo sequence. The narrative often gives rationales (quite sensible ones) for why the authors thought certain modifications would be tolerated, but getting the expected result for a single variant or two provides very thin support that rationale was correct.

Response: Again, we have now expanded our analyses to include three distinct endogenous target sites, and our results and conclusions hold overall. This helps with the concern expressed by the reviewer. In addition, most of our modification patterns lie within constant regions of the RNAs; the sequence will only change from one target site to another for 20 out of the 103 combined nts in our dual-guide system. That said, we have added a statement acknowledging the possibility that certain sequences could have differential effects on modification sensitivity.

4. Why aren't the results for C1 and C3 shown in Fig. 2A? They were not re-done in this experiment vs. the one shown in Fig. 1C? The results for C0 is a bit different quantitatively in Fig. 1C and Fig. 2A. What factors governed the reproducibility of the mCherry signal from the TLR system that this study relies so heavily on?

Response: The variation in the results for C0 between Figures 1C and 2A is not significant. Using unpaired t-test, we obtain a p value of 0.21. Therefore, the differences observed are not statistically significant. The variation observed between these experiments could be due to a number of factors such as slight differences between Cas9 purifications, cell passage number, and many others.

Reviewers' comments:

Reviewer #1 (Remarks to the Author):

I appreciate the authors response to my suggested revisions.

However, once again the reader needs to understand more about the relative purity of the oligonucleotides that were tested. What you have done is include the UV chromatogram and the deconvoluted ion chromatogram. My suggestion is/was that you give a brief table where you give the percent full-length material and tell how you calculated or determined the percent full-length material for each sequence. There are many ways to do this, I will leave it to the authors discretion to choose a method and apply it consistently. Here is the point that reader should understand: When you incorporate modifications like 2'-O-Methyl, 2'-F, and phosphorothioates into a chemically synthesized RNA, in general the more that are incorporated the more difficult it becomes to purify the oligonucleotide and that ends up effecting the overall purity that can be achieved. Heavily modified RNAs generally have a lower percentage of full-length material and that means that are other things in there that can affect your biological assays. We find that commercial research RNAs that are above 60 nucleotides in length tend be 50% to 60% full-length by ion-exchange chromatography. That means that the other 40% to 50%. At some point you decided that the purification you performed on your RNA sequences was good enough to use them in your biological assay. All I am asking is how did you make that decision? Was there a threshold where you decided that you would re-synthesize or re-purify? That is important in understanding how to interpret the biological results. I would think that you have all that data it is just a matter of putting it into a simple table. At that point you don't have to include all the mass specs.

Now in response to:

Response: We thank the reviewer for raising this important point. We take the quality control of our compounds very seriously and have now included the LC/MS chromatograms and deconvoluted mass spectrum in the supporting information confirming the M- is the expected product.

Additionally, compounds with this modification are regularly used in clinical candidates and have not shown a depyrimidination issue. Furthermore, in previous work we have carried out on 2'-fluorinated oligonucleotides (PMID 19590787), we did not see significant degradation with the conditions used for deprotection of these compounds (we tested methylamine up to several days). We did see some degradation under more strongly basic conditions (e.g. 1M sodium hydroxide at 65 °C for 8-20h).

I find your assertions a bit peculiar looking at glaringly obvious the depyrimidination peaks in your deconvoluted mass spectra; many of which you have labeled? The depyrimidination of 2'-F uridine and to a lesser extent 2'-O-Methyl uridine by strong nucleophiles such as methyl amine is well-known and well-accepted in the field and there is usually a specification set for GMP material used in clinical trials. I first heard Dr. Richard Griffey from ISIS give a talk about it in 1996, but he had the mechanism incorrect. I have included a slide I was given by Dr. Ken Hill from Agilent Technologies from a talk he gave at TIDES in 2010 where he correctly elucidated the mechanism. The Michael Addition first opens the ring yielding a urea that shows up as a peak at M-52, and then further nucleophilic attack results in M-94. The details are clearly outside the scope of your paper but if you examine the deconvoluted mass spectra it is obvious that your oligos have significantly different purities. C-2 and C-3 appear to have quite good purity but then they fall off. In C-6 you label the depyrimidination peak (M-52) which you seem to argue you have never seen (or identified).

In summary: Adding the 15 minute UV chromatogram from the LC/MS and the deconvoluted is not really helpful to the reader. A 15 minute chromatogram doesn't separate the material well enough to allow the reader to see what is in the peak and the deconvoluted mass spectra are not fully labeled to be able to see the difference between n-1, salt adducts, depurination, depyrimidination, ibu left on G etc. One example is that there is a huge difference in the apparent purity of C-8 and C-9 based on the mass spec data shown in your supplemental material. If I am going to evaluate your biological data I want to know a relative purity difference between the two oligos. Please replace the all the LC/MS data with a simple table showing a percent full-length material.

Everything else is well done and acceptable.

Reviewer #2 (Remarks to the Author):

The manuscript is significantly improved, and appropriate for publication, particularly but the comparisons of C0 and C10 modification schemes, paired with multiple modified tracrns, for additional targets HTT, HBB, and VEGFA. These data demonstrate that the same modifications to guides that preserved effectiveness with a guide targeting a particular site in GFP were also tolerated for guides of different sequence targeting different sites. These results will be helpful to the field.

I appreciate that different delivery modes, lipid vs. electroporation-based, may have different sensitivities to modifications. It is important then to make sure that results from lipid-based delivery be clearly distinguished and that seems clear enough now. Other changes to the text addressed reviewer concerns and clarifies the implications of the results.

A couple related remaining minor points / suggestions are:

1. Given that the initial rounds of modification designs relied entirely on the TRL assay system, it might be helpful to provide at least some direct assessment on reproducibility of this assay in your implementation in this study. While the experiments include 3 technical replicates of each trial, these were done together in parallel, and therefore <not> highly independent correct? It would be helpful to show a comparison (in Supp) of independent replicates (which you already have for certain reference cases) to help show the quantitative reproducibility of the assay. This issue is less critical given your validation with other guide sequences against other targets it may not be essential, but since you do use the TRL assay alone for many of your assessments of modification effect, it would be helpful to show such a comparison.
2. With only 3 replicates underlying the data, I suggest that it's better to simply show the actual 3 points for each case rather than show nearly as many (2) statistical parameters (mean and standard deviation). On the other hand, if the intention of displaying the standard deviation is to give a sense of uncertainty limits (due only to technical reproducibility of parallel replicates, then poorly defined standard deviations based on only 3 points are not a good choice of what to show. Confidence intervals (e.g. 95% CI) for example would be more appropriate for this purpose and would be based on a student t distribution.

Mir et al., NCOMMS-18-05832A ("Heavily and fully modified RNAs guide efficient SpyCas9-mediated genome editing")

Response to Critiques

We are again very grateful to the reviewers for their time and insight into our manuscript and the work that it described. The comments and criticisms have enabled us to improve the manuscript significantly. Our responses are detailed below in red.

Reviewers' comments:

Response to Reviewer #1:

I appreciate the authors response to my suggested revisions. However, once again the reader needs to understand more about the relative purity of the oligonucleotides that were tested. What you have done is include the UV chromatogram and the deconvoluted ion chromatogram. My suggestion is/was that you give a brief table where you give the percent full-length material and tell how you calculated or determined the percent full-length material for each sequence. There are many ways to do this, I will leave it to the authors discretion to choose a method and apply it consistently.

Response: We would first like to thank the reviewer for these thoughtful and constructive comments. We have now included a summary table with percent purity of each of our synthesized oligonucleotides. We have included both the percent purity by RP HPLC and by mass spectrometry, separately, for all sequences.

We see the logic of using mass spectrometry for this purity analysis (as recommended by the reviewer in his or her first set of comments) since the HPLC does not always resolve all the peaks. That said, mass spectrometry is not perfectly quantitative. For example, changes in the ionization settings can affect how much of the full-length peak appears as the molecular ion vs. as higher peaks (typically water and ion adducts). As such, to respond to the reviewer's request for a table estimating purity, we first adjusted our ionization settings to optimize further the accuracy of our results measuring content and purity. This was done by varying the voltage and temperature of the source ion spray. We then re-ran all of our oligomers using these settings.

As an example, here is the spectrum of tracrRNA T2 using our previous MS settings:

And below is the spectrum of tracrRNA T2 after optimization of the voltage and temperature increase at ion source:

No other changes were made between these two samples (i.e. the actual purity of the sample was the same in both cases, only the ionization settings were changed).

Therefore, we have added two estimates of the purity in the requested supplemental table of purity analysis: the estimate from integrating the reverse-phase LC trace, and a second estimate from integrating the deconvoluted MS trace using our optimized ionization conditions. Neither of these methods is perfect, so we are including information from both methods to ensure that we are giving our readers all the required tools to meaningfully interpret our data.

Here is the point that reader should understand: When you incorporate modifications like 2'-O-Methyl, 2'-F, and phosphorothioates into a chemically synthesized RNA, in general the more that are incorporated the more difficult it becomes to purify the oligonucleotide and that ends up effecting the overall purity that can be achieved. Heavily modified RNAs generally have a lower percentage of full-length material and that means that are other things in there that can affect your biological assays.

Response: We appreciate the reviewer's concerns about modified oligonucleotides, and we recognize the importance of the point. As described above, we have included the new supplementary table to provide the purity-related information requested in the critique.

We find that commercial research RNAs that are above 60 nucleotides in length tend to be 50% to 60% full-length by ion-exchange chromatography. That means that the other 40% to 50%. At some point you decided that the purification you performed on your RNA sequences was good enough to use them in your biological assay. All I am asking is how did you make that decision? Was there a threshold where you decided that you would re-synthesize or re-purify? That is important in understanding how to interpret the biological results. I would think that you have all that data it is just a matter of putting it into a simple table. At that point you don't have to include all the mass specs.

Response: The reviewer's questions regarding purity cutoffs (how we decide when an oligonucleotide is sufficiently pure vs. when we need to go back and re-synthesize or re-purify) are well justified, and we acknowledge the value of describing these criteria more comprehensively.

If we see (either by analytical HPLC, mass spectrometry, or dimethoxytrityl removal signal on the oligonucleotide synthesizer) that one of the monomers did not couple well, that compound is re-synthesized. If there is truncated oligo in the crude sample that is greater than ~10% of the full-length product, that means that something went wrong with either the instrument, one of the phosphoramidites or columns, or the solid support. The end result is that the instruments are then checked and

maintenance procedures are applied and the run is repeated. Over the years we have developed robust methods and procedures to synthesize diverse, highly modified, unique phosphoramidites produced by the nucleotide chemists in the Watts and Khvorova laboratories. We have extensive experience with these modified nucleotides and have specialized synthesis methods for each depending on length and types of modifications used. Unlike in a commercial setting, we optimize our instruments to use greater-than-necessary reagent concentrations to ensure that the final product is made with the highest possible yield/purity at each coupling step, rather than being concerned about reagent consumption and associated costs.

The crude samples after deprotection are then analyzed by LC/MS in-house and assessed for any failures as mentioned above. If no truncated sequences are present (including N-1) above ~10%, we consider the synthesis to have been generally successful, and the crude compound is then purified by HPLC (either RP or ion-exchange depending on the chemical modifications present on the oligonucleotide). The main peak is collected with cut-off of the peak shoulders (often everything below 80% of the peak height) and the pooled fractions are then re-analyzed by mass spectroscopy. If there is no single peak (UV or mass) that differs from our expected mass above 5-10% of the main peak, the sample is deemed pure enough for biological testing in our laboratory.

Now in response to:

Response: We thank the reviewer for raising this important point. We take the quality control of our compounds very seriously and have now included the LC/MS chromatograms and deconvoluted mass spectrum in the supporting information confirming the M- is the expected product. Additionally, compounds with this modification are regularly used in clinical candidates and have not shown a depyrimidination issue. Furthermore, in previous work we have carried out on 2'-fluorinated oligonucleotides (PMID 19590787), we did not see significant degradation with the conditions used for deprotection of these compounds (we tested methylamine up to several days). We did see some degradation under more strongly basic conditions (e.g. 1M sodium hydroxide at 65 °C for 8-20h).

I find your assertions a bit peculiar looking at glaringly obvious the depyrimidination peaks in your deconvoluted mass spectra; many of which you have labeled? The depyrimidination of 2'-F uridine and to a lesser extent 2'-O-Methyl uridine by strong nucleophiles such as methyl amine is well-known and well-accepted in the field and there is usually a specification set for GMP material used in clinical trials. I first heard Dr. Richard Griffey from ISIS give a talk about it in 1996, but he had the mechanism incorrect. I have included a slide I was given by Dr. Ken Hill from Agilent Technologies from a talk he gave at TIDES in 2010 where he correctly elucidated the mechanism. The Michael Addition first opens the ring yielding a urea that shows up as a peak at M-52, and then further nucleophilic attack results in M-94. The details are clearly outside the scope of your paper but if you examine the deconvoluted mass spectra it is obvious that your oligos have significantly different purities. C-2 and C-3 appear to have quite good purity but then they fall off. In C-6 you label the depyrimidination peak (M-52) which you seem to argue you have never seen (or identified).

Response: We thank the reviewer for this extremely useful and constructive comment, and for attaching the slide identifying the M-52 and M-94 peaks as being due to the reactivity of methylamine with 2'-F-U. To address this well-taken point, In the table of purities discussed above, we have now added comments (to the new supplementary table of purities noted above) regarding specific non-product peaks observed, including these depyrimidination peaks at M-52 and M-94, to ensure that our readers have the benefit of as much pertinent information as possible.

In summary: Adding the 15 minute UV chromatogram from the LC/MS and the deconvoluted is not really helpful to the reader. A 15 minute chromatogram doesn't separate the material well enough to allow the reader to see what is in the peak and the deconvoluted mass spectra are not fully labeled to be able to see the difference between n-1, salt adducts, depurination, depyrimidination, ibu left on G etc. One example is that there is a huge difference in the apparent purity of C-8 and C-9 based upon the mass spec data shown in your supplemental material. If I am going to evaluate your biological data I want to know a relative purity difference between the two oligos. Please replace the all the LC/MS data with a simple table showing a percent full-length material.

Response: As requested by the reviewer, and as discussed above, we have replaced all the LC/MS data with a table showing percent full-length material calculated by two methods (integrating the LC and integrating the MS).

Everything else is well done and acceptable.

Response: We thank the reviewer for this supportive comment, and for the constructive critiques that have helped us improve the manuscript. In light of the data we've added, we hope that our revised manuscript is now suitable for publication.

Reviewer #2 (Remarks to the Author):

The manuscript is significantly improved, and appropriate for publication, particularly but the comparisons of C0 and C10 modification schemes, paired with multiple modified tracers, for additional targets HTT, HBB, and VEGFA. These data demonstrate that the same modifications to guides that preserved effectiveness with a guide targeting a particular site in GFP were also tolerated for guides of different sequence targeting different sites. These results will be helpful to the field.

I appreciate that different delivery modes, lipid vs. electroporation-based, may have different sensitivities to modifications. It is important then to make sure that results from lipid-based delivery be clearly distinguished and that seems clear enough now. Other changes to the text addressed reviewer concerns and clarifies the implications of the results.

A couple related remaining minor points / suggestions are:

1. Given that the initial rounds of modification designs relied entirely on the TLR assay system, it might be helpful to provide at least some direct assessment on reproducibility of this assay in your implementation in this study. While the experiments include 3 technical replicates of each trial, these were done together in parallel, and therefore <not> highly independent correct? It would be helpful to show a comparison (in Supp) of independent replicates (which you already have for certain reference cases) to help show the quantitative reproducibility of the assay. This issue is less critical given your validation with other guide sequences against other targets it may not be essential, but since you do use the TLR assay alone for many of your assessments of modification effect, it would be helpful to show such a comparison.

Response: We thank the reviewer for this important point. The replicates shown in this study came from different populations of cells that were electroporated at different times (albeit within 48 hrs of each other). Therefore, these are independent biological replicates. However, variability resulting from different batches of purified Cas9 protein and other such factors may not be truly accounted for using these replicates.

2. With only 3 replicates underlying the data, I suggest that it's better to simply show the actual 3 points for each case rather than show nearly as many (2) statistical parameters (mean and standard deviation). On the other hand, if the intention of displaying the standard deviation is to give a sense of uncertainty limits (due only to technical reproducibility of parallel replicates, then poorly defined standard deviations based on only 3 points are not a good choice of what to show. Confidence intervals (e.g. 95% CI) for example would be more appropriate for this purpose and would be based on a student t distribution.

Response: We thank the reviewer for this comment. We have accepted the reviewer's suggestion and have now included the individual replicate points in our bar graphs to more accurately show the variability resulting from ~3 replicates.

REVIEWERS' COMMENTS:

Reviewer #1 (Remarks to the Author):

I appreciate the authors edits and addition of a table of purities.

I recommend the manuscript for publication.

Thank you

Mir et al., NCOMMS-18-05832B ("Heavily and fully modified RNAs guide efficient SpyCas9-mediated genome editing")

Response to Critiques

We are again very grateful to the reviewers for their time and insight into our manuscript and the work that it described. The comments and criticisms have enabled us to improve the manuscript significantly. Our response is detailed below in red.

Reviewers' comments:

Response to Reviewer #1:

I appreciate the authors edits and addition of a table of purities.
I recommend the manuscript for publication.

Response: Thank you!